ecology

apparent competition, boreal, woodland caribou, food webs, global change, path analysis

**Author for correspondence:**
Robert Serrouya
e-mail: serrouya@ualberta.ca

# Trophic consequences of terrestrial eutrophication for a threatened ungulate

Robert Serrouya[1], Melanie Dickie[1], Clayton Lamb[3], Harry van Oort[4], Allicia P. Kelly[5], Craig DeMars[1], Philip D. McLoughlin[6], Nicholas C. Larter[7], Dave Hervieux[8], Adam T. Ford[3] and Stan Boutin[2]

[1]Alberta Biodiversity Monitoring Institute, and [2]Department of Biological Sciences, University of Alberta, Edmonton, Alberta, Canada T6G 2E9
[3]Department of Biology, University of British Columbia, Kelowna, British Columbia, Canada V1V 1V7
[4]Environment, BC Hydro, Revelstoke, British Columbia, Canada V0E 2S0
[5]Department of Environment and Natural Resources, Government of the Northwest Territories, Fort Smith, Northwest Territories, Canada X0E 0P0
[6]Department of Biology, University of Saskatchewan, 112 Science Place, Saskatoon, Saskatchewan, Canada S7N 5E2
[7]Department of Environment and Natural Resources, Government of the Northwest Territories, Fort Simpson, Northwest Territories, Canada X0E 0N0
[8]Alberta Environment and Parks, Box 23 Provincial Building, Grande Prairie, Alberta, Canada T8V 6J4

RS, 0000-0001-5233-6081; MD, 0000-0003-2177-2352; CL, 0000-0002-1961-0509; APK, 0000-0003-0030-8913; CD, 0000-0001-7984-633X; ATF, 0000-0003-2509-7980; SB, 0000-0001-6317-038X

Changes in primary productivity have the potential to substantially alter food webs, with positive outcomes for some species and negative outcomes for others. Understanding the environmental context and species traits that give rise to these divergent outcomes is a major challenge to the generality of both theoretical and applied ecology. In aquatic systems, nutrient-mediated eutrophication has led to major declines in species diversity, motivating us to seek terrestrial analogues using a large-mammal system across 598 000 km$^2$ of the Canadian boreal forest. These forests are undergoing some of the most rapid rates of land-use change on Earth and are home to declining caribou (*Rangifer tarandus caribou*) populations. Using satellite-derived estimates of primary productivity, coupled with estimates of moose (*Alces alces*) and wolf (*Canis lupus*) abundance, we used path analyses to discriminate among hypotheses explaining how habitat alteration can affect caribou population growth. Hypotheses included food limitation, resource dominance by moose over caribou, and apparent competition with predators shared between moose and caribou. Results support apparent competition and yield estimates of wolf densities (1.8 individuals 1000 km$^{-2}$) above which caribou populations decline. Our multi-trophic analysis provides insight into the cascading effects of habitat alteration from forest cutting that destabilize terrestrial predator–prey dynamics. Finally, the path analysis highlights why conservation actions directed at the proximate cause of caribou decline have been more successful in the near term than those directed further along the trophic chain.

## 1. Introduction

Processes that increase the energy or flow of nutrients through an ecosystem can substantially alter the structure and composition of food webs [1]. At a global scale, climate change and landscape transformation are having a combined, enriching effect on ecosystems and making the world more productive [2]. However, biological diversity increases with primary productivity at first and then declines at higher levels of enrichment [3], giving rise to a 'hump-shaped' relationship between diversity and productivity [4]. In aquatic systems, increased primary productivity is referred to as eutrophication, and is often associated with

a loss of biological diversity. Eutrophication can eventually push the downward trajectory of diversity far enough to cause ecosystem collapse [5–7]. While the negative consequences for aquatic systems and plant communities are well established [5–8], it is much less clear how eutrophication impacts terrestrial food webs dominated by large-mammalian herbivores and carnivores.

One of the ways in which eutrophication affects food webs is through changes in consumer–resource interactions, including competition and predation. Competition, in the context of increasing productivity, typically explains why some species dominate or are excluded from systems with limiting resources [9]. Here, dominant competitors monopolize access to resources for sub-dominant species through exploitative or interference interactions, particularly when a species is able to persist at lower resource availability than co-occurring species (*sensu* the R* rule [9]). In some cases, increasingly favourable abiotic conditions (i.e. fluxes of nutrients or energy) can intensify competitive interactions (i.e. the stress-gradient hypothesis [10]).

In addition to affecting resource competition, increased productivity may also lead to asymmetrical predation among two prey via their shared predator. This process is termed apparent competition [11] and occurs because one prey has a higher intrinsic growth rate, with negative effects of predators 'spilling over' onto a secondary prey [11,12]. In systems where predator abundance is decoupled from the abundance of secondary prey, the secondary prey can be driven to extinction [13]. The secondary prey is a weaker competitor because it cannot persist at the same rate of predation as the primary prey, i.e. the secondary prey has a higher P*, analogous to the R* of resource-mediated competition [14].

The impact of eutrophication on species interactions within terrestrial food webs has critical implications for the management of endangered species. The boreal forests of North America are among the world's largest biomes and are undergoing rapid rates of land-use change, second only to the tropics [15]. Climate change is also accelerating in the northern latitudes of the boreal. These forests are home to woodland caribou (*Rangifer tarandus caribou*), a sub-species that is listed as threatened in Canada and is undergoing rapid range contraction. Several caribou subpopulations in southern Canada and the contiguous United States were extirpated in recent years [16], and there has been a continued recession of their southern range boundary [17,18].

While habitat alteration is a leading source of biodiversity loss globally [19,20] and for caribou specifically [18,21–23], the ultimate link between caribou declines and primary productivity is less clear. The proximate cause of caribou decline—related to lower adult survival and calf recruitment—is typically associated with predation from wolves (*Canis lupus*) and other large carnivores [24–28]. However, a paradox exists as to why caribou may not benefit from the heightened productivity of early seral forage, as do numerous other herbivores [29–34]. Caribou may be declining because of the transformation of mature forest stands into more productive early seral forage. By removing the tree canopy through forest harvesting, greatly increased levels of sunlight provide the conditions needed for understory plants to thrive [35]. These plants provide forage that benefit primary prey such as moose (*Alces alces*), deer (*Odocoileus* sp.) and their predators [36]. These predators then have a spillover effect on the naturally rarer caribou [25], which have a lower intrinsic rate of increase relative to moose [37,38], giving rise to apparent competition between moose and caribou.

The pathway of habitat alteration leading to increased productivity that favours primary prey and shared predators is complex. Although studies have evaluated individual linkages along this trophic chain [24,26,34,39,40], a more fulsome approach is to track abundance at each trophic level simultaneously. This approach also allows for explicit contrasts among a suite of potential mechanisms. For example, habitat alteration increases the vagility of predators [41] by creating movement corridors associated with forest clearing that can augment predator foraging efficiency, leading to declines in prey species [28,39]. Similarly, habitat alteration can remove winter forage for caribou, who specialize on lichens that are lost when forested stands are harvested. And finally, increased early seral forage through forest clearing may in fact benefit both caribou and moose, as has been shown with other ungulates [42]. Clearly, the potential mechanisms are diverse, and clarifying the relative support among such pathways will be important to implement evidence-based management.

Here, we contrasted direct effects of habitat alteration to effects mediated by increases in primary productivity, using the response of two primary consumers—moose and caribou—and their shared predator, wolves. We evaluated support for alternate hypotheses that potentially affected caribou population growth rates ($\lambda$): (i) habitat alteration—where resource extraction has removed forage, leading to food limitation and lower growth rates in caribou (*sensu* [24]), or (ii) habitat alteration that facilitated predator vagility, which can lead to increased foraging efficiency of wolves on caribou [41]; (iii) resource dominance—where higher productivity led to a numerical increase in moose, and consequently lower caribou $\lambda$ because caribou are a weaker competitor for a shared and limiting resource [9,43]; (iv) apparent competition—where productivity is positively linked to moose and therefore wolf abundance [44], leading to reduced caribou $\lambda$ [45]; (v) increased productivity—where both moose and caribou benefit from increased forage [29,33].

To tackle these hypotheses, we considered landscape-scale variation in primary productivity, moose and wolf densities, and caribou $\lambda$ across a 598 000-km$^2$ study area. Although other studies have quantified linkages across some of these trophic levels [24,34,45], our design attempts to capture each trophic level simultaneously across a broad gradient of habitat alteration and productivity, while explicitly contrasting alternative hypotheses. From an applied perspective, this design also allowed us to validate a previously determined threshold for wolf abundance leading to dynamically stable caribou populations [46], which has informed recovery policies for woodland caribou [47].

## 2. Methods

### (a) Experimental design and analysis

Our design was based on a mensurative experiment [48] where we estimated response metrics across a broad a range of habitat alteration, given existing landscape conditions. Habitat alteration generally decreases with increasing latitude, which introduces a potential confound with climate and by extension, primary productivity. However, jurisdictional boundaries with differing policies on resource management can provide contrasts, even at similar latitude (*sensu* [49]). In our study system, boundaries between Saskatchewan and Alberta, as well as between British Columbia (BC) and Northwest Territories (NT), provide responses from similar latitudes but vastly different amounts of habitat alteration (figure 1). These contrasts partially decouple potential confounds between habitat alteration and latitude.

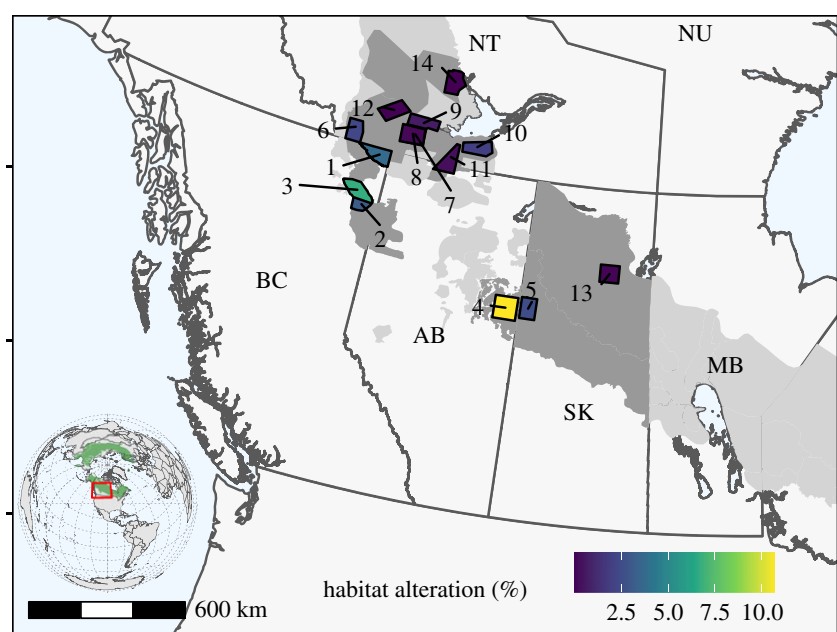

**Figure 1.** Wolf survey units (WSU) used to evaluate the relationship between productivity, moose density, wolf density and caribou population growth rates in the boreal forest of western Canada. The colour gradient represent the per cent anthropogenic habitat alteration. Light grey shading is boreal caribou range, and dark grey shading represents focal areas where caribou demographic data were collected. Numbers for each WSU correspond to the raw data labels in GitHub (https://github.com/ctlamb/borealcaribou-pathanalysis/blob/master/data/final.csv). (Online version in colour.)

To establish the link between habitat alteration and primary productivity, we first quantified a time-series relationship between forestry cutting units and the Enhanced Vegetation Index (EVI; [50]), which was our metric of primary productivity. Subsequently, we used a path analysis that encompassed three trophic levels: EVI, herbivores (i.e. moose density and caribou $\lambda$), and wolf density. We did not explicitly link habitat alteration to primary productivity in the path analysis because this would be an oversimplification of the multiple interacting factors that affect primary productivity. Landcover type and geographical location were overriding components affecting changes in primary productivity across the study area (electronic supplementary material, appendix S2), even though habitat alteration and productivity were positively correlated ($r = 0.64$, electronic supplementary material, figure S1). By contrast, there is stronger empirical support to test direct paths among vegetation, moose, wolves and caribou [41,44,46,51,52].

The path analysis was conducted across the 598 000-km$^2$ study area located in the boreal shield and boreal plains of western Canada, where 14 survey units were sampled for the abundance of wolves (wolf survey units (WSUs); figure 1). These WSUs were overlaid onto polygons where moose densities and population $\lambda$ of caribou were estimated separately as part of provincial monitoring programmes. WSU boundaries were also used to estimate primary productivity, and ranged in size from 3441 to 7266 km$^2$ (mean = 4940 km$^2$).

## (b) Environmental data
### (i) Habitat alteration
We estimated the proportion of each WSU covered by anthropogenic habitat alteration using 2008 to 2010 30-m resolution LANDSAT imagery interpreted by [47]. Linear disturbances were collected as polylines and buffered by 20 m [41]. Polygonal disturbances, such as wellsites and forestry cut blocks were then incorporated to estimate the total area directly altered by anthropogenic features. While 30-m imagery resolution does not capture smaller habitat alteration features, such as seismic exploration lines, they consistently estimate habitat alteration across the entire region of interest [53].

### (ii) Vegetation—change in Enhanced Vegetation Index (primary productivity)
We sourced a MODIS 500-m EVI product between 2000 and 2018 from https://developers.google.com/earth-engine/data-sets/catalog/MODIS_006_MOD13A1. We calculated the primary productivity of each WSU using the change in Enhanced Vegetation Index ($\Delta$EVI) from summer (when deciduous growth is peak green: 1 July–1 August) to autumn (when deciduous growth has browned, or fallen off, but snow is not common: 1 September–1 October) for each year, which isolated deciduous growth [50]. We accessed and manipulated these data using Google Earth Engine. We averaged $\Delta$EVI values across the five years previous to the year of the moose census bounded by the WSU, because $\Delta$EVI fluctuates annually and there are spatial gaps in the satellite coverage.

## (c) Demographic data
### (i) Moose
We used available population density estimates to quantify the moose density within areas overlapping the WSUs. Aerial moose surveys were conducted by provincial governments, academic or industry partners between 2008 and 2018 (electronic supplementary material, appendix S3). In some cases, WSUs overlapped multiple moose survey polygons, in which case we calculated density based on the weighted average of each polygon's area. Moose survey data were not available in southern Saskatchewan. We, therefore, estimated the density of moose using remote wildlife cameras, and corrected camera densities to aerial survey densities using a correlation analysis conducted on camera and aerial survey data collected across Alberta (electronic supplementary material, figure S3-1).

### (ii) Wolves
Wolf densities were estimated in WSUs using aerial surveys conducted from 2015 to 2020. Two WSUs had considerable spatial overlap, but wolves were surveyed in separate years (2016 and 2017). We considered these two WSUs as independent samples. Aerial surveys were designed based on an *a priori* power analysis that sought to optimize aerial transect spacing to maximize the

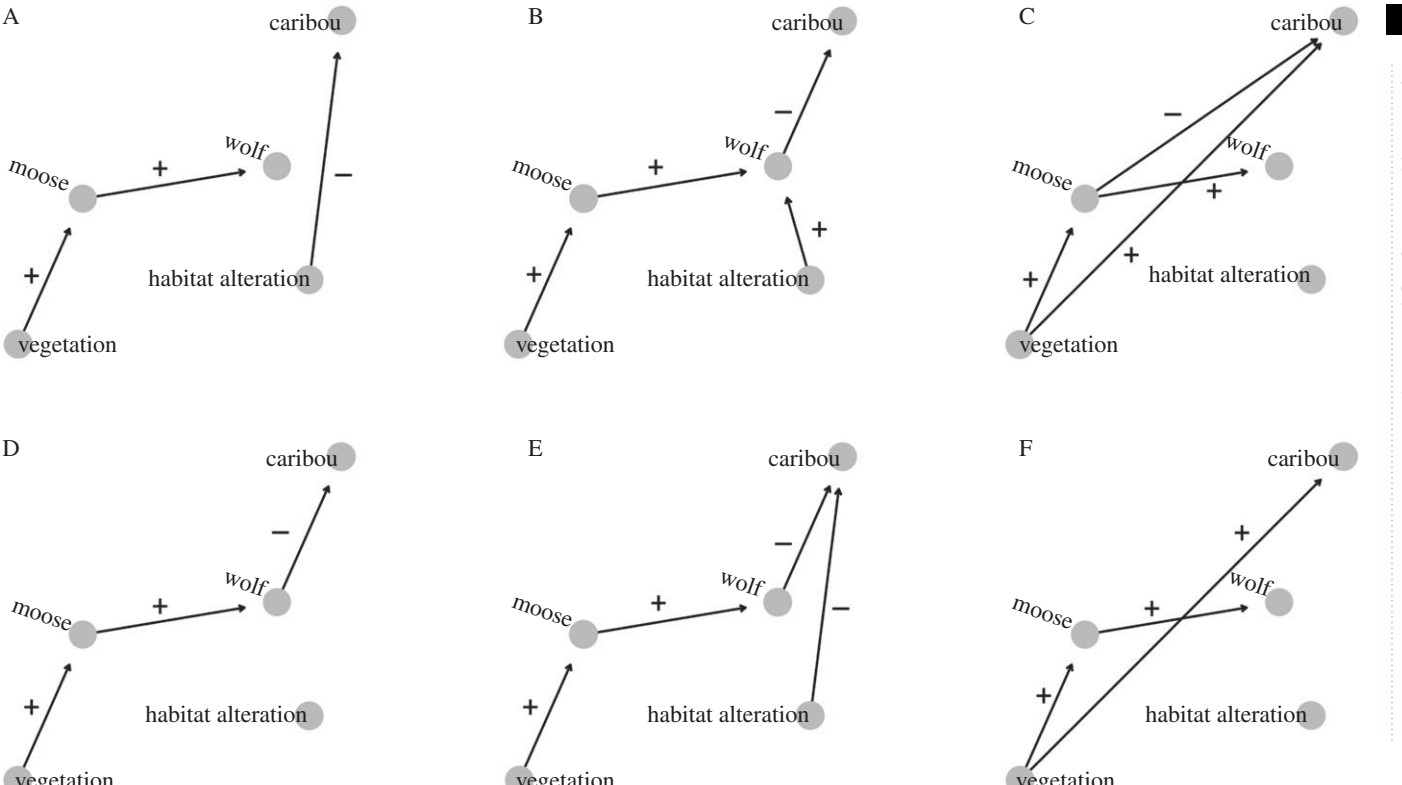

**Figure 2.** Directed acyclic graphs displaying six hypothesized ecological pathways impacting caribou $\lambda$. A, Habitat alteration leading to food limitation; B, habitat alteration leading to increased predator vagility; C, resource dominance of moose over caribou; D, apparent competition; E, apparent competition combined with food limitation; F, increased productivity benefitting caribou and moose. Vegetation represents our estimate of primary productivity ($\Delta$EVI), moose and wolves are densities, caribou refers to $\lambda$, and habitat alteration is the percentage area covered by anthropogenic habitat alteration. For (C, D and F), the link from habitat alteration to vegetation was estimated as part of separate analyses (figure 3 and electronic supplementary material, appendix S2).

detection of wolves and their tracks in snow. A full description of survey methodology is provided in the electronic supplementary material, appendix S4, but can be summarized as using empirical wolf movement data based on 5-min locations from [41] coupled with simulations to clarify the trade-off between how wide transect spacing could be in relation to how much a wolf travels over a given window of time. Based on these analyses, the sampling intensity was set at 3-km spacing (electronic supplementary material, figure S4-1).

### (iii) Caribou

We obtained caribou demographic data published from provincial and territorial monitoring programmes, updated with additional estimates that overlap our study period. Estimates of annual adult female survival, calf recruitment and the number of individuals monitored to obtain these estimates were collected from the following caribou ranges: East Side of the Athabasca River, Cold Lake (Alberta), Cold Lake (Saskatchewan), Saskatchewan boreal shield, Calendar, Chinchaga (British Columbia), Clarke, Hay River Lowlands, Pine Point-Buffalo Lake, South Dehcho, North Dehcho and Yates. We used recruitment and survival rates to estimate the finite population growth rate $\lambda$ [54], using the equation of [55] adjusted by [56] which accounts for the delayed age at first parturition of caribou. The equation is $\lambda = S/(1 - R)$, $R = (X/2)/(1 + (X/2))$, where $S =$ adult female survival, $R =$ recruitment, $X =$ the ratio of juveniles per adult female. This adjustment is algebraically identical to a Lefkovitch stage matrix with three stages [56], and is the convention for boreal caribou population monitoring in much of Canada [57]. We calculated the geometric mean $\lambda$ from three years preceding each wolf survey, to account for the fact that juvenile recruitment can be highly variable from year to year [58], and a multi-year average is more likely to represent landscape conditions.

## (d) Analyses
### (i) Habitat alteration influence on primary productivity

We assessed the influence of habitat alteration on primary productivity within 2956 forestry cut blocks that were greater than 0.25 km$^2$, which also is the resolution of the primary productivity data ($\Delta$EVI). We conducted this analysis across 598 000 km$^2$, defined by the 100% minimum convex polygon of the 14 WSUs. The vegetation index was at a 500-m resolution, thus we focused on logging-related habitat alteration, as cut blocks are appropriately sized for a 500-m pixel to measure changes, whereas linear disturbances (most are less than 40-m wide) were not appropriate for this analysis. We used cut blocks logged between 2003 and 2016, which provided at least three years of pre- and post-alteration data in which the 2000–2019 vegetation index data were available. For each cut block we calculated a pre-disturbance baseline value, and calculated the per cent divergence of all vegetation index values (ranging from –10 to 16 years relative to alteration date) from this baseline value.

### (ii) Path analysis

We used path analysis to examine the causal relationships between primary productivity ($\Delta$EVI), moose density, wolf density and caribou $\lambda$. Path analyses are appropriate when independent variables are simultaneously treated as dependent factors within one investigation. Hypothesized pathways influencing caribou $\lambda$ are shown in the directed acyclic graph (DAG; figure 2), including: food limitation, where anthropogenic habitat alteration lowers caribou carrying capacity [24] (figure 2a); increased predator vagility—where habitat alteration increases wolf movement rates leading to increased predation and a lower caribou $\lambda$ [41,59] (figure 2b); resource dominance—where vegetation limits both caribou and moose abundance,

but moose, the stronger competitor, exert a negative influence on caribou $\lambda$ (figure 2c); apparent competition—where wolves are supported by moose and exert a negative effect on caribou $\lambda$ (figure 2d); a combined effect—where both food limitation and apparent competition occur (figure 2e); and direct enrichment—where caribou populations benefit from increased primary productivity (figure 2f) (sensu [29]).

We linear-transformed variables where appropriate to reduce model complexity, and ensured that any transformations created linear relationships among variables of interest along the paths. We used directional separation (D-separation) to falsify our multivariate causal hypotheses [60] following the approach of [61]. Using the steps of D-separation, we first determined a set of independent claims that were required to be true for the structure of the hypothesized DAG to be correct. Second, we calculated a p-value associated with each of these claims using linear models. As an example, consider the causal relationship where A causes B which causes C, $A \rightarrow B \rightarrow C$. An independent claim required for this structure to be true would be that A has no significant effect on C, after controlling for the effect of B. The test of this would be to create a linear model of form $C \sim A + B$, where p of A must be $p > 0.05$ for the claim to be supported. Third, we used these probabilities to calculate Fisher's C statistic ($-2\Sigma \ln(P)$), which follows a $\chi^2$ distribution with $2k$ degrees of freedom, to calculate a single metric for the D-separation test. A D-separation test with a p-value $\leq 0.05$ indicates that the proposed correlation structure of the model differs from that observed in the data, and the DAG is, therefore, rejected. As suggested by [61], causal models which were not rejected were compared using Akaike's information criterion (AIC) [62] corrected for small sample sizes (AICc). The direction and strength of relationships for top-supported pathways were calculated by averaging standardized coefficients from competing models ($\Delta AICc < 3$) by their model weight. Finally, we used 1000 bootstrapped samples and the D-separation approach to plot the relative support for each pathway. For each bootstrapped sample we retained the top model ($\Delta AICc = 0$) and plotted the path.

### (iii) Threshold for stable caribou populations

Bergerud & Elliot [46] proposed that caribou recruitment would offset mortality up to a density of 6.5 wolves 1000 $km^{-2}$. This threshold was estimated by plotting the relationship of both caribou mortality and recruitment as a function of wolf density, and the intersection of these two curves corresponded to 6.5 wolves 1000 $km^{-2}$. Using similar reasoning, we estimated the wolf density at which caribou $\lambda = 1$, by intersecting the y-intercept at 1 with the predicted relationship between wolf abundance and caribou $\lambda$, which is equivalent to a value of recruitment offsetting mortality. Furthermore, we extrapolated this approach to moose, by estimating the moose density at which caribou $\lambda = 1$. Regressions were bootstrapped ($n = 1000$) to estimate the uncertainty of these thresholds.

## 3. Results

### (a) Habitat alteration influence on primary productivity

Across 2956 forestry cut blocks (average = 0.62 $km^2$ (95% confidence interval (CI): 0.26–1.64)) we detected a consistent pattern of increased $\Delta EVI$ following habitat alteration. $\Delta EVI$ decreased immediately after alteration, but within a year values exceed pre-alteration values, and these effects persisted at least 16 years (the maximum time we were able to assess these data). $\Delta EVI$ values remain 15–45% higher than pre-alteration values 2–16 years post-alteration (figure 3).

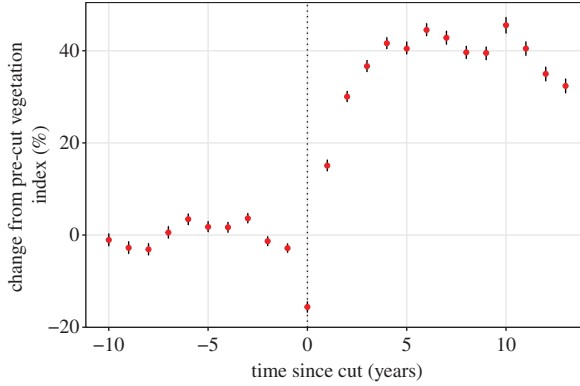

**Figure 3.** Vegetation index ($\Delta EVI$) time series before and after habitat alteration from forest harvesting. Vertical line denotes the year of alteration. Error bars show 95% CIs based on 500 bootstrapped samples. A mixed model with each cut block as the random slope predicts 35.6% (s.e. = 0.27) greater $\Delta EVI$ post logging. (Online version in colour.)

### (b) Path analysis

Hypotheses A, C and F differed significantly ($p = 0.036$, 0.040 and 0.011, respectively) from the observed data, and were rejected. Hypotheses B, D and E did not differ significantly ($p = 0.433$, 0.509 and 0.429, respectively) from the observed data. Comparing these latter three models using AICc (AICc: B = 140.42, D = 102.32 and E = 140.61) revealed no support for B or E ($\Delta AICc = 38.1$ and 38.3, respectively) [63], and the data provide much stronger support for hypothesis D (apparent competition) as the top model (table 1). Furthermore, hypotheses D had nine parameters, as did hypotheses A and F, but the latter two were greater than 35 AICc units higher (https://github.com/ctlamb/borealcaribou-pathanalysis) suggesting that hypothesis D was the top model, not simply because it had the fewest parameters. The $R^2$ from linear models for each step in model D were generally high: moose $\sim$ vegetation = 0.43, wolf $\sim$ moose = 0.77, caribou $\sim$ wolf = 0.71 (figure 4). Bootstrapped samples were congruent with support for model D, which was selected as the top model in 82% of samples (figure 5). The value at which caribou population growth is stable ($\lambda = 1$) is 1.8 (95% CI: 0.8–2.91) wolves 1000 $km^{-2}$ and 29.0 (19.2–36.6) moose 1000 $km^{-2}$.

## 4. Discussion

Worldwide increases in primary productivity [2] are expected to have positive effects on ecosystem services such as favourable outcomes for agricultural production [64,65], or increased potential for wood fibre as a consequence of expanding treelines [66,67]. Many of these benefits, however, are likely to be partially offset by drought, wildfire, pests and disease, highlighting the difficulty of anticipating alternative states under global greening [65,68]. Similarly, the uncertainty of putative outcomes in ecological communities will increase with food webs that are complex, particularly if they are governed by indirect interactions such as apparent competition. Theoretical underpinnings will clarify the range of possible outcomes [14,69,70], but only field studies can evaluate the veracity of these general predictions [71]. Such experiments will be particularly difficult to implement in large-mammal systems, but our mensurative study contributes to this growing body of empirical work [72,73]. In particular, we found that higher levels of primary productivity reduced the ability of a

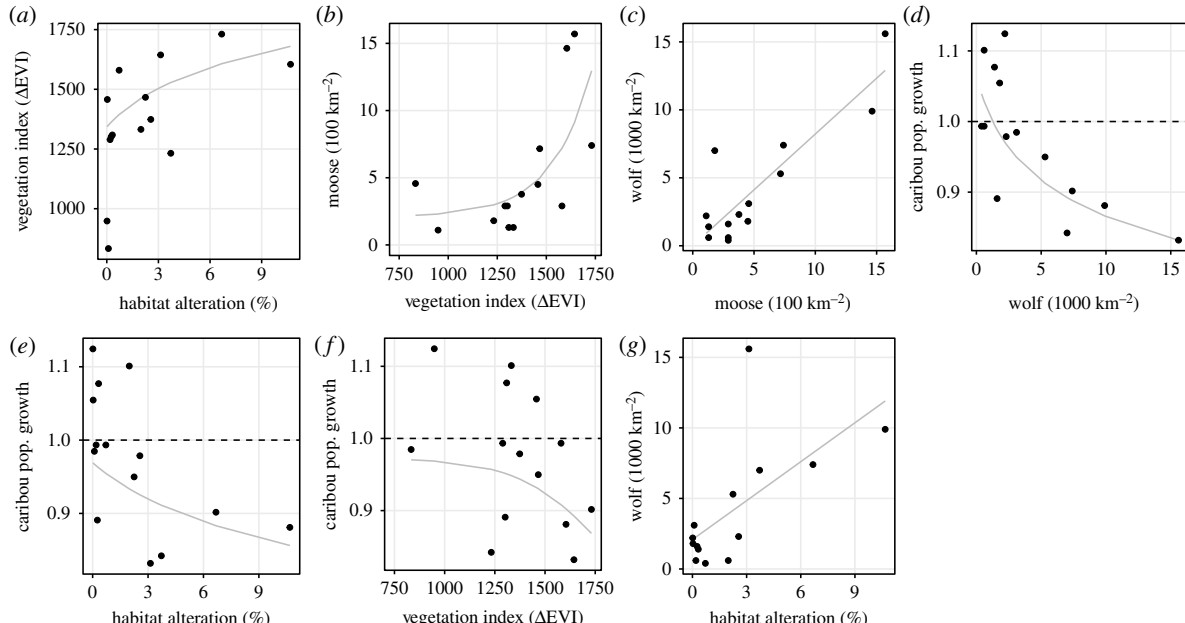

**Figure 4.** Bivariate relationships included in the path analyses (figure 2). Note that the top performing model includes panels (*b*), (*c*) and (*d*) (table 1 and figure 5) which describe apparent competition (model D, figure 2). Caribou population growth refers to $\lambda$ values, and the horizontal dashed line represents a stable population where $\lambda = 1$. Pairwise relationships among all variables are in the electronic supplementary material, figure S1-1.

**Table 1.** Results of the path analyses from models presented in figure 2. (D-separation tests were used to identify model fit to the data. *p*-values (less than 0.05) indicate whether the proposed correlation structure of the model differs from the data, and if so, the model was rejected. Models not rejected with *p*-values were compared using AIC corrected for small sample size (AICc). *K* is the number of parameters.)

| model | hypothesis | K | p | AICc | ΔAICc |
|---|---|---|---|---|---|
| D | apparent competition | 9 | 0.509 | 102.3 | 0.00 |
| B | habitat alteration: predator vagility | 10 | 0.433 | 140.4 | 38.1 |
| E | apparent competition and food limitation | 10 | 0.429 | 140.6 | 38.3 |

weaker large-mammal competitor to persist, which is the mechanism that has been invoked to explain the hump-shaped pattern observed in broader studies of biodiversity [3]. That caribou have a lower intrinsic growth rate relative to moose [37,38], potentially driven by a more constrained foraging niche [74], makes them susceptible to landscape changes that favour the abundance of other ungulates.

For organisms like woodland caribou, which are adapted to nutrient-poor environments, the effects of increased productivity are expected to be predominantly negative [75–77], though at least two mechanisms can influence this pattern. The first involves the direct loss of resources, with a comparatively simple pathway explained by the recession of lower productivity habitats that are replaced by more productive, greener ecosystems [78]. This mechanism has been observed from taxa as diverse as butterflies affected by rising treeline [67,79], birds that specialize in acidic bogs that become converted to shrubs [80] and herbivores whose preferred grasslands are replaced by forests [72]. This mechanism, which we represented as habitat loss leading to food limitation (model A), was not supported for woodland caribou in our study. Similarly, habitat change leading to increased predator foraging efficiency was not the most parsimonious explanation of our data (model B).

The second suite of mechanisms invoke relatively indirect pathways such as competitive interactions including resource dominance leading to exploitative competition, or the complex inter-trophic pathway of apparent competition. Theory allows us to predict that in simple cases of exploitative competition, if carrying capacity is increased owing to increased resources, all else being equal, the competitor with the higher intrinsic growth rate (a component of being a stronger competitor) will dominate [9]. This mechanism was also not supported in our study, even though moose have a higher intrinsic growth rate, with greater niche breadth [37,38,74]. Moose did not appear to exert competitive exclusion on caribou, probably because these species do not have a major overlap of needed resources [74,81].

For caribou, the results strongly support a complex multi-trophic pathway described as apparent competition. However, this finding highlights a paradox, because [34] found that forest stands with higher productivity recovered more quickly following disturbance, leading to reduced impacts on caribou. Yet our findings indicate that if the mean primary productivity increases across large spatial scales, thereby increasing the carrying capacity of primary prey, outcomes will be predominantly negative for caribou and other victim species of apparent competition. Independent theoretical [45] and empirical evidence [21,26] support this general prediction.

Global analyses of eutrophication [2] can mask local dynamics of productivity, even for large biomes such as the boreal forest. Factors influencing variation in productivity are

**Figure 5.** Results of the path analysis used to explain ecosystem dynamics. (*a*) Relative support for each pathway based on 1000 bootstrapped samples, top path ($\Delta$AICc = 0) retained from each run. (*b*) Strength ($R^2$ shown along the path, with standardized coefficients in brackets) and direction of relationships for top model identified in (*a*). The dashed line represents a link estimated as part of separate analyses (figure 3 and electronic supplementary material, appendix S2). (Online version in colour.)

complex, many of which will be heavily influenced by natural components such as latitude, elevation and underlying edaphic properties. Nonetheless, we have shown that humans can influence productivity based on the time-series analysis of forest stands regenerating from harvest, with mounting evidence that this process is occurring in the boreal forest [40]. A likely compounding factor that we did not directly estimate in our study is atmospheric $CO_2$ fertilization, a global phenomenon that is predicted to exacerbate the eutrophication of the boreal forest biome [2].

Canada's recovery strategy for boreal woodland caribou is based on a national-scale meta-analysis that produced a robust relationship between increased habitat alteration and reduced caribou vital rates [47,82]. This relationship guides federal policy intended to increase habitat conservation and restoration with the goal of achieving self-sustaining woodland caribou populations across Canada. The relationship between habitat alteration and caribou demography was not intended to be mechanistic because there is incomplete information on other ecosystem components (prey and predator densities) at the national scale. While the spatial extent of our study was more restricted than this national analysis, it afforded the opportunity to contrast mechanistic pathways driving caribou population growth, with stronger relationships identified among wolves, moose and caribou, compared to habitat alteration directly influencing caribou. We note that our analysis did not explicitly address wildfire, which is a widespread disturbance in the boreal forest, yet $\Delta$EVI would implicitly incorporate the effects of wildfire [83]. In addition, [82] showed that wildfire had much less influence on caribou demography relative to human-caused habitat alteration (and see [84]). We also found that [46]'s threshold of 6.5 wolves $1000\ km^{-2}$ and a more conservative federal target of 3.0 $1000\ km^{-2}$ [85] are both too high for our study system (which is restricted to the boreal forest), where a density up to 1.8 wolves $1000\ km^{-2}$ is more likely to yield stable caribou populations. Extending the analysis to moose densities, 29.0 $1000\ km^{-2}$ was the threshold where caribou populations tended to decline, which may not be compatible with maximum sustained-yield harvest approaches to moose management [86].

Recovering species affected by habitat-mediated apparent competition requires combinations of actions that include reducing invading prey species, reducing predators and reversing habitat alteration that led to increased prey [87]. Our results highlight why conservation actions directed

closer to the proximate cause of decline (predation; [16]) may have been more successful in expeditiously growing caribou populations in the short term than those directed further along the trophic chain, such as prey reductions [86], or reducing habitat productivity via restoration [88]. The relationship between wolf density and caribou $\lambda$ displayed the largest effect size compared to relationships at lower trophic levels. Nonetheless, addressing ultimate factors contributing to apparent competition, caused by increased primary productivity from anthropogenic habitat alteration, is self-evidently an important objective to achieve woodland caribou recovery. Moreover, the ability to study each trophic level simultaneously across a range of habitat alteration presents a compelling case of how increasing productivity can have cascading effects contributing to the extirpation of species maladapted to terrestrial eutrophication.

**Ethics.** No animals were handled as part of this study. Caribou demographic rates were obtained from regular provincial and territorial monitoring programmes, as were moose surveys. Aerial wolf surveys were conducted by the authors and permits were: Government of Alberta (permit no. 57102, collection licence 57103), Government of Saskatchewan (17FW008) and Government of the Northwest Territories (WL500382 and WL500383).

**Authors' contributions.** R.S. and H.V.O. conceived the study. C.L. carried out the statistical analyses. R.S., A.T.F., M.D., C.L., S.B., P.D.M., A.P.K., D.H., C.D. and H.V.O. wrote the manuscript. M.D. and H.V.O. coordinated the study. All authors critically revised the manuscript and gave final approval for publication and agree to be held accountable for the work performed therein.

**Data accessibility.** Raw code and data for all analyses are presented on GitHub (https://github.com/ctlamb/borealcaribou-pathanalysis).

**Competing interests.** We declare we have no competing interests.

**Funding.** Government of the Northwest Territories, Government of Alberta, Resource Industry Caribou Collaboration, British Columbia Oil, and Gas Research and Innovation Society funded this study.

**Acknowledgements.** We are grateful to our First Nations partners for their support of boreal caribou monitoring in BC and NT: Fort Nelson First Nation, Prophet River First Nation, Sambaa K'e Dene Band, Fort Simpson Métis Local, The Denendeh Harvesters Committee of Líídlíí Kue First Nation, Jean Marie River First Nation, Pehdzeh Ki First Nation, Nahanni Butte Dene Band, Acho Dene Koe Band, Deh Gah Got'ie First Nation, Fort Providence Métis Council, Ka'a'gee Tu First Nation, West Point First Nation, K'atl'odeeche First Nation, Deninu Kue First Nation and the Northwest Territory Métis Nation. We sincerely thank Derek Drinnan (Black Sheep Aviation) and Glen Watts who assisted with wolf surveys, and to Kendal Benesh, Scott McNay, and Steve Wilson who provided logistical support.

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
