## [Reviewer comments · Proceedings of the Royal Society B: Biological Sciences]

Review History

RSPB-2020-1250.R0 (Original submission)

Review form: Reviewer 1

Recommendation

Major revision is needed (please make suggestions in comments)

Scientific importance: Is the manuscript an original and important contribution to its field?

Acceptable

General interest: Is the paper of sufficient general interest?

Excellent

Quality of the paper: Is the overall quality of the paper suitable?

Good

Is the length of the paper justified?

Yes

Should the paper be seen by a specialist statistical reviewer?

No

Do you have any concerns about statistical analyses in this paper? If so, please specify them explicitly in your report.

Yes

It is a condition of publication that authors make their supporting data, code and materials available - either as supplementary material or hosted in an external repository. Please rate, if applicable, the supporting data on the following criteria.

Is it accessible?

Yes

Is it clear?

Yes

Is it adequate?

Yes

Do you have any ethical concerns with this paper?

No

Comments to the Author

Major comments

1. I have followed the fantastic work by this group of researchers over the past two decades. To my mind, it is one of the most compelling examples of apparent competition in a large mammal study system. That stated, and after reading the Introduction, I found myself wondering whether and how this particular study offered novel insights into the wolf-moose-caribou dynamic. I understand that the use of path analysis to disentangle various hypotheses may be a new analytical approach, but this seems like a thin hair to slice: aren't we already reasonably confident that apparent competition between moose and caribou – ultimately driven by forest cutting, proximately driven by wolf predation – underlies declines of woodland caribou? In short, and from a macro perspective, this study overlaps heavily with those already published by this group.

That stated, I am less familiar with this study system than some, and I am obviously less familiar with this study system than the authors. And I recognize that most science is incremental, and this kind of work requires many years of successive study (and publication).

Although it may not be perfectly clear to most readers, the distinction between this study and past work may be perfectly clear to the authors. So, I believe that they should be given an opportunity to more clearly distinguish the results – not the methods or statistical approach, necessarily – of the current study from, e.g., Wittmer et al 2005 *Oecologia*, Wittmer et al 2007 *J Animal Ecol*, Hervieux et al 2007 *CJZ*, Serrouya et al 2015 *Am Nat*, Serrouya et al 2019 *PNAS*, etc. They might start by identifying one or more knowledge gaps in the study system that the current study addresses.

2. Although the trend for increasing primary productivity following habitat alteration is clear, it looks like there is not a significant difference between pre and post habitat alteration (Fig. 3). Error bars in Figure 3 would probably show considerable overlap if these contain (more typical) 95% CIs instead of (somewhat atypical) 33-66 % quantiles. This could muddy the authors' conclusion that habitat alteration negatively affects caribou by enhancing primary productivity (L419-420).

3. L404-405, L419-421: Along with major comment #2, I'd suggest that the authors discuss the need to establish a firmer mechanistic link between habitat alteration and primary

productivity, if indeed this wasn't achieved in the current study.

4. Hypothesis #1 (food limitation; L101-102; L239-240) states that habitat alteration leads to food limitation through logging and other disturbances that reduce food availability for caribou. However, Hypothesis #3 (resource dominance; L241-143; L104-106) states that this habitat alteration enriches vegetation (as shown in results), thereby increasing food availability, for which moose are claimed to be a better competitor than caribou. If moose are 'better competitors', then in the absence of moose, caribou should benefit from habitat alteration that enhances primary productivity. But this is contrary to the premise of Hypothesis #1--that habitat alteration leads to food limitation. In sum, it seems that habitat alteration can lead to both food limitation or food enrichment (in the absence of moose) for caribou. The food limitation hypothesis makes more sense when the authors discuss that, despite vegetation enrichment, unique resources could be lost (L354-364).

5. Table 1: Shouldn't the AICc model comparison exclude the models unsupported by the D-separation test, resulting in a comparison of only models D, E, and B? This might change the relative weights between these models.

Minor comments

1. L27: When first 'weaker competitors' is invoked, please define it operationally. Although nearly all readers will be familiar with this term, its definition will vary among them.
2. L30: I believe that 'Earth' should be capitalized.
3. L33: if 'ecosystem enrichment' = 'increased productivity', please use just one of these terms for consistency. If they are intended to mean different things, please state how they differ.
4. L35: See minor comment #1, and apply it to 'resource dominance'.
5. L68: Please articulate how 'complex food web' differs from 'food web', or delete 'complex'.
6. L70: 'or lower vulnerability to predation' can be deleted, since apparent competition typically will not occur if a more vulnerable prey also exhibits a higher intrinsic growth rate.
7. L79: here and throughout, I believe that 'species' should be 'subspecies'.
8. L84-85: see above wrt 'ecosystem enrichment'.
9. L88-89: suggest simply using 'herbivores' instead of 'small herbivores and large ungulates' (since all North American ungulates are herbivores).
10. L89: 'enrichment via' can be deleted.
11. L92-93: one or more citations are needed to support this claim.
12. L95-97: this is a jargon-rich sentence. Is it possible to more simply state exactly what you mean here? For example, 'forest clearing' is more intuitive than 'resource extraction'.
13. L101-102: I'm confused. Up until this point, the implication has been that logging/forest clearing/resource extraction/industrial activity enhances/enriches/increases productivity/food; is this known, hypothesized, or assumed? Also, and for each combination of terms here, please pick one and use it consistently, or distinguish these terms from each other.
14. L154-163: was habitat alteration calculated for one year, for separate years, or summarized across years? Did the time period of habitat alteration data overlap the time period from which demographic data was used?
15. L172-173: "We averaged Δ EVI values across the five years previous to the year of the moose census bounded by the WSU". Was there any particular reason for using this five-year criterion?
16. L212: for those who do not have the time or inclination to consult the original references (i.e., most readers), please provide the original equation and adjustment here.
17. L215-216: similar to minor comment #15, why was caribou mean lambda calculated from 3 years preceding each wolf survey?
18. L336-337: please provide more details on how these thresholds were obtained.
19. L351: along the same lines as minor comment #1, what is the evidence that caribou are competitively subordinate to moose? Apologies if I missed this earlier in the text. Regardless, please state this or remind readers of it here. This is mentioned off-handedly in L369-370, but this should come when first dominant/subordinate competitors are mentioned.

20. L401: "the Δ EVI values would implicitly incorporate effects of wildfire". If there's no citation for this, some explanation would be helpful.

Review form: Reviewer 2

Recommendation

Accept with minor revision (please list in comments)

Scientific importance: Is the manuscript an original and important contribution to its field?

Excellent

General interest: Is the paper of sufficient general interest?

Excellent

Quality of the paper: Is the overall quality of the paper suitable?

Excellent

Is the length of the paper justified?

Yes

Should the paper be seen by a specialist statistical reviewer?

No

Do you have any concerns about statistical analyses in this paper? If so, please specify them explicitly in your report.

No

It is a condition of publication that authors make their supporting data, code and materials available - either as supplementary material or hosted in an external repository. Please rate, if applicable, the supporting data on the following criteria.

Is it accessible?

Yes

Is it clear?

Yes

Is it adequate?

Yes

Do you have any ethical concerns with this paper?

No

Comments to the Author

GENERAL SUMMARY - see attached file for further detailed comments

The manuscript presents an investigation of how eutrophication, i.e. increase in primary productivity, affected a terrestrial food web dominated by large mammal herbivores and carnivores. Five hypotheses for how primary productivity and habitat alteration may affect competition and predation amongst caribou, moose, and wolves in the boreal forests of Canada. Specifically, "we contrasted direct effects of habitat alteration compared to effects mediated by increases in primary productivity, using the response of two primary consumers – moose and

caribou – and their shared predator, wolves.”

These boreal forest systems are indeed rapidly changing and the focal species of caribou is notably threatened in Canada. I found the manuscript well-written, enjoyable to read, and the hypotheses tested in this system interesting in general. My review is concentrated on aspects related to the Methods section, mostly with some areas that would benefit from clarification. I have one major comment on the choice to exclude the link between habitat alteration and vegetation (primary productivity) from the path analysis, particularly because it is the primary stated direct/indirect effect being examined. Overall, I think my comments constitute a minor revision.

Review form: Reviewer 3 (J Fryxell)

Recommendation

Accept with minor revision (please list in comments)

Scientific importance: Is the manuscript an original and important contribution to its field?

Excellent

General interest: Is the paper of sufficient general interest?

Excellent

Quality of the paper: Is the overall quality of the paper suitable?

Excellent

Is the length of the paper justified?

Yes

Should the paper be seen by a specialist statistical reviewer?

No

Do you have any concerns about statistical analyses in this paper? If so, please specify them explicitly in your report.

No

It is a condition of publication that authors make their supporting data, code and materials available - either as supplementary material or hosted in an external repository. Please rate, if applicable, the supporting data on the following criteria.

Is it accessible?

Yes

Is it clear?

Yes

Is it adequate?

Yes

Do you have any ethical concerns with this paper?

No

Comments to the Author

This MS describes a coarse comparison of green vegetation abundance, forest disturbance level, moose density, wolf density, and estimated rate of woodland caribou population change, to test 5 general hypotheses that might be expected to apply when extensive forest clearing occurs. This question is highly relevant in both a general sense to community and ecosystem ecologists and in a conservation sense because woodland caribou are declining across much of their boreal forest range in Canada. Data for wildlife management units were pulled across a wide spatial range. Path analysis was used to discriminate among competing models, with results clearly favoring apparent competition competition between moose and caribou, due to elevated levels of wolf predation, as the most plausible model/hypothesis. The paper has many strengths, foremost being its application of data from multiple spatial locations, yielding wide variation in disturbance levels. Since many of the alternative models involve involve complex causal networks, path analysis is clearly the best (and perhaps only) way to treat such snapshot data. The presentation is clear and well supported by accompanying tables and figures.

I have only 1 major complaint. Given the small number of data points that go into a crucial analysis, I think readers ought to see all the data displayed in a table and additional subplots added to display the visual evidence the crucial relationships that are responsible for rejecting alternate models. For example, there seems little support for the effect of disturbance per se on caribou lambda. Great - let's see it, so we can assess whether outliers, nonlinearities, or other data features cloud the simple statistical inference.

I also think that eutrophication is perhaps a misleading term, because enrichment is not usually interpreted as being synonymous with forest clearing. This semantic issue will no doubt irritate many aquatic ecologists and is not necessary for the paper to be of major importance. I would hate to see its value lost in a silly debate about terminology.

John Fryxell
University of Guelph

Decision letter (RSPB-2020-1250.R0)

10-Jul-2020

Dear Dr Serrouya:

I am writing to inform you that your manuscript RSPB-2020-1250 entitled "Trophic consequences of terrestrial eutrophication for a threatened ungulate" has, in its current form, been rejected for publication in Proceedings B.

This action has been taken on the advice of referees, who have recommended that substantial revisions are necessary. With this in mind we would be happy to consider a resubmission, provided the comments of the referees are fully addressed. However please note that this is not a provisional acceptance.

Sincerely,
 Dr Sasha Dall
 mailto: proceedingsb@royalsociety.org

Associate Editor
 Board Member: 1
 Comments to Author:

Following the completion of three excellent reviews, it is apparent the reviewers are enthusiastic about the manuscript and consider it of high importance if some significant concerns can be addressed. In particular, please address carefully the first major concern raised by reviewer 1: how is this current manuscript distinct from the previous excellent research that has been completed in this study system? Also, all three reviewers raise important points concerning the analysis, presentation and interpretation of the data, and these points should each be addressed as well.

Reviewer(s)' Comments to Author:

Referee: 1

Comments to the Author(s)

Major comments

1. I have followed the fantastic work by this group of researchers over the past two decades. To my mind, it is one of the most compelling examples of apparent competition in a large mammal study system. That stated, and after reading the Introduction, I found myself wondering whether and how this particular study offered novel insights into the wolf-moose-caribou dynamic. I understand that the use of path analysis to disentangle various hypotheses may be a new analytical approach, but this seems like a thin hair to slice: aren't we already reasonably confident that apparent competition between moose and caribou – ultimately driven by forest cutting, proximately driven by wolf predation – underlies declines of woodland caribou? In short, and from a macro perspective, this study overlaps heavily with those already published by this group.

That stated, I am less familiar with this study system than some, and I am obviously less familiar with this study system than the authors. And I recognize that most science is incremental, and this kind of work requires many years of successive study (and publication).

Although it may not be perfectly clear to most readers, the distinction between this study and past work may be perfectly clear to the authors. So, I believe that they should be given an opportunity to more clearly distinguish the results – not the methods or statistical approach, necessarily – of the current study from, e.g., Wittmer et al 2005 *Oecologia*, Wittmer et al 2007 *J Animal Ecol*, Hervieux et al 2007 *CJZ*, Serrouya et al 2015 *Am Nat*, Serrouya et al 2019 *PNAS*, etc.

They might start by identifying one or more knowledge gaps in the study system that the current study addresses.

2. Although the trend for increasing primary productivity following habitat alteration is clear, it looks like there is not a significant difference between pre and post habitat alteration (Fig. 3). Error bars in Figure 3 would probably show considerable overlap if these contain (more typical) 95% CIs instead of (somewhat atypical) 33-66 % quantiles. This could muddy the authors' conclusion that habitat alteration negatively affects caribou by enhancing primary productivity (L419-420).

3. L404-405, L419-421: Along with major comment #2, I'd suggest that the authors discuss the need to establish a firmer mechanistic link between habitat alteration and primary productivity, if indeed this wasn't achieved in the current study.

4. Hypothesis #1 (food limitation; L101-102; L239-240) states that habitat alteration leads to food limitation through logging and other disturbances that reduce food availability for caribou. However, Hypothesis #3 (resource dominance; L241-143; L104-106) states that this habitat alteration enriches vegetation (as shown in results), thereby increasing food availability, for which moose are claimed to be a better competitor than caribou. If moose are 'better competitors', then in the absence of moose, caribou should benefit from habitat alteration that enhances primary productivity. But this is contrary to the premise of Hypothesis #1--that habitat alteration leads to food limitation. In sum, it seems that habitat alteration can lead to both food limitation or food enrichment (in the absence of moose) for caribou. The food limitation hypothesis makes more sense when the authors discuss that, despite vegetation enrichment, unique resources could be lost (L354-364).

5. Table 1: Shouldn't the AICc model comparison exclude the models unsupported by the D-separation test, resulting in a comparison of only models D, E, and B? This might change the relative weights between these models.

Minor comments

1. L27: When first 'weaker competitors' is invoked, please define it operationally. Although nearly all readers will be familiar with this term, its definition will vary among them.

2. L30: I believe that 'Earth' should be capitalized.

3. L33: if 'ecosystem enrichment' = 'increased productivity', please use just one of these terms for consistency. If they are intended to mean different things, please state how they differ.

4. L35: See minor comment #1, and apply it to 'resource dominance'.

5. L68: Please articulate how 'complex food web' differs from 'food web', or delete 'complex'.

6. L70: 'or lower vulnerability to predation' can be deleted, since apparent competition typically will not occur if a more vulnerable prey also exhibits a higher intrinsic growth rate.

7. L79: here and throughout, I believe that 'species' should be 'subspecies'.

8. L84-85: see above wrt 'ecosystem enrichment'.

9. L88-89: suggest simply using 'herbivores' instead of 'small herbivores and large ungulates' (since all North American ungulates are herbivores).

10. L89: 'enrichment via' can be deleted.

11. L92-93: one or more citations are needed to support this claim.

12. L95-97: this is a jargon-rich sentence. Is it possible to more simply state exactly what you mean here? For example, 'forest clearing' is more intuitive than 'resource extraction'.

13. L101-102: I'm confused. Up until this point, the implication has been that logging/forest clearing/resource extraction/industrial activity enhances/enriches/increases productivity/food; is this known, hypothesized, or assumed? Also, and for each combination of terms here, please pick one and use it consistently, or distinguish these terms from each other.

14. L154-163: was habitat alteration calculated for one year, for separate years, or summarized across years? Did the time period of habitat alteration data overlap the time period from which demographic data was used?

15. L172-173: "We averaged Δ EVI values across the five years previous to the year of the moose census bounded by the WSU". Was there any particular reason for using this five-previous-years criterion?
16. L212: for those who do not have the time or inclination to consult the original references (i.e., most readers), please provide the original equation and adjustment here.
17. L215-216: similar to minor comment #15, why was caribou mean lambda calculated from 3 years preceding each wolf survey?
18. L336-337: please provide more details on how these thresholds were obtained.
19. L351: along the same lines as minor comment #1, what is the evidence that caribou are competitively subordinate to moose? Apologies if I missed this earlier in the text. Regardless, please state this or remind readers of it here. This is mentioned off-handedly in L369-370, but this should come when first dominant/subordinate competitors are mentioned.
20. L401: "the Δ EVI values would implicitly incorporate effects of wildfire". If there's no citation for this, some explanation would be helpful.

Referee: 2

Comments to the Author(s)

GENERAL SUMMARY - see attached file for further detailed comments

The manuscript presents an investigation of how eutrophication, i.e. increase in primary productivity, affected a terrestrial food web dominated by large mammal herbivores and carnivores. Five hypotheses for how primary productivity and habitat alteration may affect competition and predation amongst caribou, moose, and wolves in the boreal forests of Canada. Specifically, "we contrasted direct effects of habitat alteration compared to effects mediated by increases in primary productivity, using the response of two primary consumers – moose and caribou – and their shared predator, wolves."

These boreal forest systems are indeed rapidly changing and the focal species of caribou is notably threatened in Canada. I found the manuscript well-written, enjoyable to read, and the hypotheses tested in this system interesting in general. My review is concentrated on aspects related to the Methods section, mostly with some areas that would benefit from clarification. I have one major comment on the choice to exclude the link between habitat alteration and vegetation (primary productivity) from the path analysis, particularly because it is the primary stated direct/indirect effect being examined. Overall, I think my comments constitute a minor revision.

Referee: 3

Comments to the Author(s)

This MS describes a coarse comparison of green vegetation abundance, forest disturbance level, moose density, wolf density, and estimated rate of woodland caribou population change, to test 5 general hypotheses that might be expected to apply when extensive forest clearing occurs. This question is highly relevant in both a general sense to community and ecosystem ecologists and in a conservation sense because woodland caribou are declining across much of their boreal forest range in Canada. Data for wildlife management units were pulled across a wide spatial range. Path analysis was used to discriminate among competing models, with results clearly favoring apparent competition between moose and caribou, due to elevated levels of wolf predation, as the most plausible model/hypothesis. The paper has many strengths, foremost being its application of data from multiple spatial locations, yielding wide variation in disturbance levels. Since many of the alternative models involve complex causal networks, path analysis is clearly the best (and perhaps only) way to treat such snapshot data. The presentation is clear and well supported by accompanying tables and figures.

I have only 1 major complaint. Given the small number of data points that go into a crucial analysis, I think readers ought to see all the data displayed in a table and additional subplots

added to display the visual evidence the crucial relationships that are responsible for rejecting alternate models. For example, there seems little support for the effect of disturbance per se on caribou lambda. Great - let's see it, so we can assess whether outliers, nonlinearities, or other data features cloud the simple statistical inference.

I also think that eutrophication is perhaps a misleading term, because enrichment is not usually interpreted as being synonymous with forest clearing. This semantic issue will no doubt irritate many aquatic ecologists and is not necessary for the paper to be of major importance. I would hate to see its value lost in a silly debate about terminology.

John Fryxell
University of Guelph

Author's Response to Decision Letter for (RSPB-2020-1250.R0)

See Appendix A.

RSPB-2020-2811.R0

Review form: Reviewer 1

Recommendation

Accept as is

Scientific importance: Is the manuscript an original and important contribution to its field?

Excellent

General interest: Is the paper of sufficient general interest?

Good

Quality of the paper: Is the overall quality of the paper suitable?

Excellent

Is the length of the paper justified?

Yes

Should the paper be seen by a specialist statistical reviewer?

No

Do you have any concerns about statistical analyses in this paper? If so, please specify them explicitly in your report.

No

It is a condition of publication that authors make their supporting data, code and materials available - either as supplementary material or hosted in an external repository. Please rate, if applicable, the supporting data on the following criteria.

Is it accessible?

Yes

Is it clear?

Yes

Is it adequate?

Yes

Do you have any ethical concerns with this paper?

No

Comments to the Author

The authors have done a good job of addressing comments. This is an excellent piece of work.

My only minor comment is that I think it's worth mentioning the time span during which satellite-quantified habitat alteration occurred.

Decision letter (RSPB-2020-2811.R0)

11-Dec-2020

Dear Dr Serrouya

I am pleased to inform you that your manuscript RSPB-2020-2811 entitled "Trophic consequences of terrestrial eutrophication for a threatened ungulate" has been accepted for publication in Proceedings B.

The referee(s) have recommended publication, but also suggest some minor revisions to your manuscript. Therefore, I invite you to respond to the comments and revise your manuscript. Because the schedule for publication is very tight, it is a condition of publication that you submit the revised version of your manuscript within 7 days. If you do not think you will be able to meet this date please let us know.

- 1) A text file of the manuscript (doc, txt, rtf or tex), including the references, tables (including captions) and figure captions. Please remove any tracked changes from the text before submission. PDF files are not an accepted format for the "Main Document".
- 2) A separate electronic file of each figure (tiff, EPS or print-quality PDF preferred). The format should be produced directly from original creation package, or original software format. PowerPoint files are not accepted.

3) Electronic supplementary material: this should be contained in a separate file and where possible, all ESM should be combined into a single file. All supplementary materials accompanying an accepted article will be treated as in their final form. They will be published alongside the paper on the journal website and posted on the online figshare repository. Files on figshare will be made available approximately one week before the accompanying article so that the supplementary material can be attributed a unique DOI.

Sincerely,

Dr Sasha Dall

Associate Editor

Board Member

Comments to Author:

I commend the authors for their thorough revision of their manuscript in response to the reviews they received. Please address the lone remaining comment from reviewer #1: "My only minor comment is that I think it's worth mentioning the time span during which satellite-quantified habitat alteration occurred."

Reviewer(s)' Comments to Author:

Referee: 1

Comments to the Author(s).

The authors have done a good job of addressing comments. This is an excellent piece of work.

My only minor comment is that I think it's worth mentioning the time span during which satellite-quantified habitat alteration occurred.

Author's Response to Decision Letter for (RSPB-2020-2811.R0)

See Appendix B.

Decision letter (RSPB-2020-2811.R1)

15-Dec-2020

Dear Dr Serrouya

I am pleased to inform you that your manuscript entitled "Trophic consequences of terrestrial eutrophication for a threatened ungulate" has been accepted for publication in Proceedings B.

Open Access

Paper charges

Sincerely,
Editor, Proceedings B
<mailto:proceedingsb@royalsociety.org>

Appendix A

10 November 2020

Dear Dr Sasha Dall and the Proceedings Editorial Board,

We wish to sincerely thank the three reviewers and the Associate Editor for their thoughtful comments on our manuscript entitled “Trophic consequences of terrestrial eutrophication for a threatened ungulate.”

We have addressed each point in detail below in *italics*. In particular, we clarified how this work is an advance over previous research on habitat-mediated apparent competition, by highlighting that all trophic levels were studied simultaneously across a broad spectrum of habitat alteration and primary productivity. Previous studies examined only 1 or 2 links in the trophic chain, or lacked explicit evidence that habitat alteration can increase primary productivity. We have also clarified how some mechanisms stemming from forest harvesting were apparently contradictory (Reviewer 1 comments), but are now more clearly stated to be tested as contrasting hypotheses.

We apologize for the delay in re-submitting the manuscript. In late winter 2020, the government of the Northwest Territories conducted four additional wolf surveys (using the same field personnel and methods), two of which contained corresponding data on caribou and moose, so we chose to include those two surveys in the analysis (based in part on the comment from Reviewer 3 regarding the limited number of data points, albeit conducted across a broad spatial scale). It took some time to amalgamate the caribou demographic data. Results were largely unaffected, with the apparent competition hypothesis still being the clear winner, although R^2 values decreased slightly.

We look forward to the next set of comments on our manuscript.

Regards,
Robert Serrouya (on behalf of all authors)

Associate Editor

Comments to Author:

Following the completion of three excellent reviews, it is apparent the reviewers are enthusiastic about the manuscript and consider it of high importance if some significant concerns can be addressed. In particular, please address carefully the first major concern raised by reviewer 1: how is this current manuscript distinct from the previous excellent research that has been completed in this study system? Also, all three reviewers raise important points concerning the analysis, presentation and interpretation of the data, and these points should each be addressed as well.

Reviewer(s)' Comments to Author:

Referee: 1

Major comments

1. I have followed the fantastic work by this group of researchers over the past two decades. To my mind, it is one of the most compelling examples of apparent competition in a large mammal study system. That stated, and after reading the Introduction, I found myself wondering whether and how this particular study offered novel insights into the wolf-moose-caribou dynamic. I understand that the use of path analysis to disentangle various hypotheses may be a new analytical approach, but this seems like a thin hair to slice: aren't we already reasonably confident that apparent competition between moose and caribou—ultimately driven by forest cutting, proximately driven by wolf predation—underlies declines of woodland caribou? In short, and from a macro perspective, this study overlaps heavily with those already published by this group.

That stated, I am less familiar with this study system than some, and I am obviously less familiar with this study system than the authors. And I recognize that most science is incremental, and this kind of work requires many years of successive study (and publication).

Although it may not be perfectly clear to most readers, the distinction between this study and past work may be perfectly clear to the authors. So, I believe that they should be given an opportunity to more clearly distinguish the results—not the methods or statistical approach, necessarily—of the current study from, e.g., Wittmer et al 2005 *Oecologia*, Wittmer et al 2007 *J Animal Ecol*, Hervieux et al 2007 *CJZ*, Serrouya et al 2015 *Am Nat*, Serrouya et al 2019 *PNAS*, etc. They might start by identifying one or more knowledge gaps in the study system that the current study addresses.

We think it is an excellent suggestion to emphasise why this study goes beyond previous examples of habitat-mediated apparent competition, in the caribou literature and more generally. We think the main advance is that abundance/trend is tracked at each trophic level, instead of inferred, or where only 2 trophic levels are tracked. A brief summary of pathways in previous work:

Seip 1992: Wolf -> Caribou, general mention of moose, no link to habitat alteration.

Wittmer Oecologia 2005: habitat -> caribou density (no primary prey, no predators),

Wittmer et al. 2007: habitat -> caribou survival (no primary prey, no predators)

Serrouya et al. 2015: primary prey -> predators -> caribou (no link of habitat to primary prey)

Serrouya et al. 2019: predator -> caribou or primary prey -> caribou (no link to habitat, or primary prey to predators to caribou)

This is why we think our new study system in the boreal forest represents the most complete picture, with explicit contrasts among alternative hypotheses.

In the Introduction we now write on L. 101 “Although studies have evaluated individual linkages along this trophic chain [26,34,39–41], a more fulsome approach is to track each trophic level simultaneously. This approach also allows for explicit contrasts among a suite of potential mechanisms.” and on L. 125 “Although other studies have quantified linkages across some of

these trophic levels [34,39,46], our design attempts to capture each trophic level simultaneously across a broad gradient of habitat alteration and productivity, while explicitly contrasting alternative hypotheses.”

2. Although the trend for increasing primary productivity following habitat alteration is clear, it looks like there is not a significant difference between pre and post habitat alteration (Fig. 3). Error bars in Figure 3 would probably show considerable overlap if these contain (more typical) 95% CIs instead of (somewhat atypical) 33-66 % quantiles. This could muddy the authors’ conclusion that habitat alteration negatively affects caribou by enhancing primary productivity (L419-420).

We now conduct a mixed model with each clearcut as a random slope and test for effects of vegetation index pre- and post-logging. Results show 35.6% (SE = 0.27) greater Vegetation index post logging. Error bars are 95% CIs (higher precision is achieved when lumping pre and post logging).

We also bootstrapped the results, by resampling with replacement the cutblocks. We then calculated a mean annual change (with a mixed model with each clearcut as a random slope) for each year. This approach better shows the statistical support for whether an increase did or did not occur in each year. Our previous approach, showing annual quantiles across all the data, was misleading because the wide intervals indicated a potential lack of statistical support. Now, the 95% CI signifies where 95% of means if re-sampled would fall within the error bars.

3. L404-405, L419-421: Along with major comment #2, I'd suggest that the authors discuss the need to establish a firmer mechanistic link between habitat alteration and primary productivity, if indeed this wasn't achieved in the current study.

We reason that the previous analysis has addressed this comment.

4. Hypothesis #1 (food limitation; L101-102; L239-240) states that habitat alteration leads to food limitation through logging and other disturbances that reduce food availability for caribou. However, Hypothesis #3 (resource dominance; L241-143; L104-106) states that this habitat alteration enriches vegetation (as shown in results), thereby increasing food availability, for which moose are claimed to be a better competitor than caribou. If moose are 'better competitors', then in the absence of moose, caribou should benefit from habitat alteration that enhances primary productivity. But this is contrary to the premise of Hypothesis #1--that habitat alteration leads to food limitation. In sum, it seems that habitat alteration can lead to both food limitation or food enrichment (in the absence of moose) for caribou. The food limitation hypothesis makes more sense when the authors discuss that, despite vegetation enrichment, unique resources could be lost (L354-364).

Thank you for pointing out this apparent contradiction. We now clarify how both mechanisms could operate - food limitation via removal of resources (trees and therefore lichens) vs tree removal that leads to enrichment via increased shrub growth. We now add the following clarification, in the paragraph preceding the hypothesis paragraph.

L. 99: The pathway of habitat alteration leading increased productivity that favours primary prey and shared predators, is complex. Although studies have evaluated individual linkages along this trophic chain [26,34,39-41], a more fulsome approach is to track each trophic level simultaneously. This approach also allows for explicit contrasts among a suite of potential mechanisms. For example, habitat alteration can substantially increase the vagility of predators [42] by creating movement corridors associated with forest clearing that can increase

predator foraging efficiency, leading to declines in prey species [40,43]. Similarly, habitat alteration can remove forage for caribou, who specialize on lichens that are lost when forested stands are harvested. And finally, increased forage through forest clearing may in fact benefit both caribou and moose, as has been shown with other ungulates [29,31,33]. Clearly, the potential mechanisms are diverse, and clarifying the relative support among such pathways will be important to implement evidence-based management responses.

We hope that this clarifies the apparent dichotomy, and given that Hypothesis #1 and #3 received little support relative to apparent competition (#4), the implications will be more clear to readers.

5. Table 1: Shouldn't the AICc model comparison exclude the models unsupported by the D-separation test, resulting in a comparison of only models D, E, and B? This might change the relative weights between these models.

We also agree with this comment and have removed the models with $p < 0.05$

Minor comments

1. L27: When first 'weaker competitors' is invoked, please define it operationally. Although nearly all readers will be familiar with this term, its definition will vary among them.

To avoid ambiguity (given limited word space) in the abstract, we rephrased the sentence with our initial reference to 'weaker competitors' to: "Changes in primary productivity have the potential to substantially alter food webs, with positive outcomes for some species and negative outcomes for others. Understanding the environmental context and species traits that give rise to these divergent outcomes is major challenge to the generality of both theoretical and applied ecology. In aquatic systems, nutrient-mediated eutrophication has led to major declines in species diversity, motivating us to seek terrestrial analogues using a large-mammal system across 598,000 km² of the Canadian boreal forest."

2. L30: I believe that 'Earth' should be capitalized.

Agreed.

3. L33: if 'ecosystem enrichment' = 'increased productivity', please use just one of these terms for consistency. If they are intended to mean different things, please state how they differ.

Agreed. We now consistently use the terms eutrophication (to draw conceptual links between our work and aquatic ecology) and primary productivity or increased productivity. We removed ecosystem enrichment, though we do use the term 'enriching' in one instance.

4. L35: See minor comment #1, and apply it to 'resource dominance'.

We changed this wording to be more specific with respect to how competition studies are often interpreted:

"In food webs that include indirect interactions, increased productivity may also lead to asymmetrical predation among two prey via their shared predator. This process is termed apparent competition [11] and occurs because one prey has a higher intrinsic growth rate (r), with negative effects of predators 'spilling over' onto a secondary prey [11,12]. In systems where predator abundance is decoupled from the abundance of secondary prey, the secondary prey can be driven to extinction [13]. The secondary prey is a weaker competitor because it cannot persist at the same rate of predation as the primary prey, i.e. the secondary prey has a higher P^ , analogous to the R^* of resource-mediated competition*

5. L68: Please articulate how 'complex food web' differs from 'food web', or delete 'complex'.

We changed this sentence to read "In food webs that include indirect interactions, increased productivity may also lead to asymmetrical predation among two prey via their shared predator."

6. L70: 'or lower vulnerability to predation' can be deleted, since apparent competition typically will not occur if a more vulnerable prey also exhibits a higher intrinsic growth rate.

Thank you.

7. L79: here and throughout, I believe that 'species' should be 'subspecies'.

Agreed.

8. L84-85: see above wrt 'ecosystem enrichment'.

We have changed to consistently use the term 'primary productivity.'

9. L88-89: suggest simply using 'herbivores' instead of 'small herbivores and large ungulates' (since all North American ungulates are herbivores).

Agreed.

10. L89: 'enrichment via' can be deleted.

Agreed

11. L92-93: one or more citations are needed to support this claim.

Potvin, F., Courtois, R., 2004. Winter presence of moose in clear-cut black spruce landscapes: related to spatial pattern or to vegetation. Alces 40, 61–70.

*Bjørneraas, K., Solberg, E.J., Herfindal, I., Moorter, B., Rolandsen, C., Tremblay, J.-P., Skarpe, C., Sæther, B., Eriksen, R., Astrup, R., 2011. Moose (*Alces alces*) habitat use at multiple temporal scales in a human-altered landscape. Wildl. Biol. 17, 44–54.*

12. L95-97: this is a jargon-rich sentence. Is it possible to more simply state exactly what you mean here? For example, 'forest clearing' is more intuitive than 'resource extraction'.

We have changed the sentence to read "Habitat alteration can also substantially increase the vagility of predators [41] by creating movement corridors associated with forest clearing that can increase predator foraging efficiency, leading to declines in prey species. [39,42]. "

13. L101-102: I'm confused. Up until this point, the implication has been that logging/forest clearing/resource extraction/industrial activity enhances/enriches/increases productivity/food; is this known, hypothesized, or assumed? Also, and for each combination of terms here, please pick one and use it consistently, or distinguish these terms from each other.

Similar to major comment #4, we think we have addressed this problem in the paragraph preceding the hypothesis statements (Lines 99-110) . We now clarify how forest clearing could provide opposing outcomes (removing lichen and/or increasing shrub growth), and how this could affect ungulates in opposing manners. We are building the case of how the path analysis can help discriminate among these competing pathways/mechanisms.

14. L154-163: was habitat alteration calculated for one year, for separate years, or summarized across years? Did the time period of habitat alteration data overlap the time period from which demographic data was used?

Habitat alteration was the total cumulative amount of area affected by human activity "as visible" from landsat imagery, quantified for the federal boreal caribou disturbance mapping initiative http://ec.gc.ca/data_donnees/STB-DGST/001/Boreal_Caribou_2011_Scientific_Assessment_Mapping_Methods_Appendix_-_ENGLISH.pdf. So the disturbance goes back several decades before the demographic data were collected, yet this historic disturbance is known to have a legacy effect because of the long time frame for natural succession to occur. We now write "Habitat alteration was delineated

using 30-m resolution LANDSAT imagery interpreted for any feature that was clearly visible at a viewing scale of 1:50 000”

15. L172-173: "We averaged ΔEVI values across the five years previous to the year of the moose census bounded by the WSU". Was there any particular reason for using this five-year previous-years criterion?

We now write “We averaged ΔEVI values across the five years previous to the year of the moose census bounded by the WSU, because ΔEVI fluctuates annually and there are spatial gaps in the satellite coverage. Moose density is unlikely to respond rapidly to annual variation in ΔEVI but more so the spatial productivity among areas.”

16. L212: for those who do not have the time or inclination to consult the original references (i.e., most readers), please provide the original equation and adjustment here.

We now insert “ The equation is $\lambda = S / (1 - R)$, $R = (X / 2) / (1 + (X / 2))$, where S = adult female survival, R = recruitment, X = the ratio of juveniles per adult female.“

17. L215-216: similar to minor comment #15, why was caribou mean λ calculated from 3 years preceding each wolf survey?

We now add “We calculated the geometric mean λ from three years preceding each wolf survey, to account for the fact that juvenile recruitment can be highly variable from year to year [62], and a multi-year average is more likely to represent landscape conditions.”

18. L336-337: please provide more details on how these thresholds were obtained.

In the Methods section we wrote “Using similar reasoning, we estimated the wolf density at which caribou $\lambda = 1$ by intersecting the y intercept of 1 with the predicted relationship between wolf abundance and caribou λ , which is equivalent to a value of recruitment offsetting mortality “, but now add on L309 “by intersecting the y-intercept of 1 with the predicted relationship between wolf abundance and caribou λ ”

19. L351: along the same lines as minor comment #1, what is the evidence that caribou are competitively subordinate to moose? Apologies if I missed this earlier in the text. Regardless, please state this or remind readers of it here. This is mentioned off-handedly in L369-370, but this should come when first dominant/subordinate competitors are mentioned.

The most direct line of evidence is that caribou have a lower intrinsic growth rate (delayed age at first parturition, almost never twin, whereas twinning is common with moose), and narrower niche breadth. We now elaborate this concept here (“That caribou have a lower r relative to moose [37,38], potentially driven by more constrained foraging niche [78], make them susceptible to landscape changes that favour the abundance of other ungulates.” but also earlier in the text (L. 97) because we neglected to do so - thank you.

20. L401: "the Δ EVI values would implicitly incorporate effects of wildfire". If there's no citation for this, some explanation would be helpful.

We added a citation here (Zheng et al. 2016).

Referee: 2

Comments to the Author(s)

GENERAL SUMMARY

The manuscript presents an investigation of how eutrophication, i.e. increase in primary productivity, affected a terrestrial food web dominated by large mammal herbivores and carnivores. Five hypotheses for how primary productivity and habitat alteration may affect competition and predation amongst caribou, moose, and wolves in the boreal forests of Canada. Specifically, “we contrasted direct effects of habitat alteration compared to effects mediated by increases in primary productivity, using the response of two primary consumers – moose and caribou – and their shared predator, wolves.”

These boreal forest systems are indeed rapidly changing and the focal species of caribou is notably threatened in Canada. I found the manuscript well-written, enjoyable to read, and the hypotheses tested in this system interesting in general. My review is concentrated on aspects related to the Methods section, mostly with some areas that would benefit from clarification. I have one major comment on the choice to exclude the link between habitat alteration and vegetation (primary productivity) from the path analysis, particularly because it is the primary stated direct/indirect effect being examined. Overall, I think my comments constitute a minor revision.

We thank the reviewer for these comments, and in particular for digging into the raw code.

Major comments

I agree that only habitat alteration as an explanatory variable of EVI is a strong simplification but I'm wondering if the comparison of the indirect vs. direct effects of habitat alteration is lost to some degree by not including that path in the hypothetical models. Some reasoning is below but I think given the hypotheses as they're laid out it is defensible to include the path from habitat to vegetation while admitting the limitation that this is certainly not the only, and often not strongest, factor affecting primary productivity in this system. While I think having the path explicit could be helpful for general interpretation, ultimately, the results and interpretation are the same with these data so perhaps the point is moot.

We struggled greatly with this concept at the outset of the paper, for two reasons. First, as stated, the relationship between habitat alteration and dEVI is much more complex than the other causal pathways (e.g. moose to wolves, wolves to caribou), because dEVI is driven by far more than just habitat alteration. This is why we analysed this component separately and in a more fulsome manner in Appendix 2. We relied on this analysis (Appendix 2), which is more causative, as well as the time-series analysis of increased dEVI post logging (Fig. 3), to establish the link between human-caused habitat alteration and increased productivity (dEVI). Even though habitat alteration and dEVI are highly correlated (Appendix 1), we feel this is partly spurious due to the latitudinal gradient covered by our study. Second, adding that extra path of Habitat Alteration to dEVI causes us to ‘run out’ of degrees of freedom, by adding 2 degrees of freedom (intercept, slope). We will now also emphasize that even though we have limited degrees of freedom, apparent competition is the top model, even when compared to other

hypotheses that have the same number of parameters used (eg Habitat alteration (Model A) and productivity increase (Model F); Fig. 1; and see L. 338), to highlight that the results are not simply an artefact having a single model with the fewest K being the winner in the information-theoretic framework.

- Land Cover type, temperature, and precipitation are always expected to be major factors of primary productivity. In your model explaining primary productivity, it appears that habitat alteration has a similar strength as precipitation but land cover and temperature are really the major drivers. (Checking That I'm understanding and interpreting this correctly)

Indeed, this interpretation is correct, we will clarify this in the caption. But a key message is that human activity has indeed moved the needle on this important metric.

- If I understood the analysis for the appendix, the results are based on one year of EVI (change-values ?) while the metric used in the path analysis is the 5-year average of change-values. The bivariate correlation between habitat disturbance and EVI seems relatively weak in the data in the appendix compared to the fairly strong correlation in the aggregated path model data. Given that these aren't exactly the same variables, is the relationship with temperature less/more/same correlated with the EVI and habitat alteration of the path model?

Yes you understand this correctly. Temperature and the other explanatory variables we use in Appendix 2 (landcover, precipitation) would be equally predictive in the aggregated path model. But because our sample size was relatively limited in the path, we were unable to create such a complex multivariate prediction of the factors influencing productivity (dEVI). Instead, we leverage a much larger dataset to show how habitat disturbance affects dEVI directly. We did this in two ways, a large spatial analysis compared dEVI values across a gradient of habitat alteration while controlling for temperature, precipitation, and landcover type (Appendix 2), and then a temporal analysis that clearly shows the effect of forest removal (logging) on increasing dEVI. We think that this approach provides a rigorous, and mechanistic link between dEVI and habitat alteration, while also making the most efficient use of data available.

- It was a bit surprising to me to see the negative marginal effect of precipitation but I'm also less familiar with the boreal forest system. Is that relationship typical or perhaps the result of the multivariate model and the correlation between temperature and precipitation? (not critical nor a shortcoming, but my curiosity)

Agreed. We believe this is partly to do with upland vs lowland habitats and water retention/water tables. Some areas with high rainfall may not always produce higher amounts of deciduous shrubs. This finding is similar to that found in

(<https://onlinelibrary.wiley.com/doi/full/10.1111/j.1365->

[2486.2008.01612.x?casa_token=IOBABjj2k1kAAAAA%3ASbk5m9IGniQ5zEJiYkUC-6SKQaeWwHkef-C1CZqa40eMeM64yF1WgHS4o8STaxjSVwYUXrzKWqbQmq](https://doi.org/10.2486/2008.01612.x?casa_token=IOBABjj2k1kAAAAA%3ASbk5m9IGniQ5zEJiYkUC-6SKQaeWwHkef-C1CZqa40eMeM64yF1WgHS4o8STaxjSVwYUXrzKWqbQmq)) where Net Ecosystem Productivity was slightly lower in areas with high precipitation.

Methods and analysis appear generally sound but there are a few things that remained unclear to me from the descriptions.

- I'm unclear on whether the 598,000 km² is referring to the total area of the WSUs or the total area of the boreal forest indicated in the map.

This area is the approximate 100% minimum convex polygon encompassing all of the WSUs, and represents the area where the time series and EVI analyses occurred. We now write "We conducted this analysis across a 598,000-km² area, defined by the 100% minimum convex polygon of the 14 WSUs"

- What is the overlap of the caribou data and the WSUs?

We now highlight on the map the specific caribou ranges where caribou data were collected, along with the full boreal caribou range as added context. Where possible we specifically highlight the focal area in which caribou demographic data were obtained; for example, see defined caribou ranges in British Columbia and Alberta. However, in the case of the Saskatchewan boreal shield and boreal plains populations, defined caribou ranges are not delineated, but the caribou data were collected in close proximity to the WSU in conjunction with the research lead in that province (P. McLoughlin, co-author of this MS).

- What is being accomplished by the bootstrapping in the context of the path analysis? Looking at the code I figured out that the data was bootstrapped 1000 times for each proposed structural model and I think making this clear in the text would be very helpful for better understanding the structural model testing and comparison - assuming I did interpret this correctly.

Yes this is correct. Although we already incorporate uncertainty into our analysis via the error structure of the linear models, we wanted to test the sensitivity of our relatively small sample to deviations in sample. So we bootstrapped the analysis 1000 times and assessed how many times we got the same answer, which gave us confidence that although small, our sample provided consistent results even when resampled.

- I didn't see results reported relating to the averaging of the standardized coefficients L263-265, with r^2 values reported in Figure 5 where I would normally expect the standardized coefficient estimates in a path model. Given that it's a set of bivariate linear models and no variables have a direct and indirect effects I think reporting the r^2 values on the paths is okay, though a little unconventional for path models. Ideally, is reporting both coefficient estimates (paths) and r^2 at the node of the response variable possible?

Thanks for the suggestion. We have now added the standardized coefficient in brackets after the R^2 . We chose to maintain the location along the path, instead of at the node so that it was clear to the reader what portion of the path the numbers were referring to.

- In the delineation of habitat alteration, was this heads up or a product of Environment and Climatechange Canada? Is the 1:50,000 scale for meters?

It was a product of ECCC that is commonly used for studies of boreal caribou demography-habitat relationships. Yes, we will clarify meters.

Minor comments

1. The map figure requires revision to address the following issues:

- It's impossible to identify the overlapping WSUs, i.e. it looks like there are 11 on the map

We clarified this with the new numbering system. Thank you.

- The water-body outlines coming from under the boreal forest layer was very confusing for a while. It Would be ideal to have those be on top.

We hope this problem is addressed in the new map.

- Add provincial labels at minimum (e.g. BC, NT), and if possible some labeling of the WSUs with corresponding description in caption to match with names in appendices.

Agreed and done.

- Is the % habitat alteration only anthropogenic disturbance or all disturbances? Add this detail to the caption

It is only anthropogenic alteration, we have adjusted the caption.

2. Is LAI in the code the same as EVI in the text? It would be helpful to reconcile the variable names.

Thank you for this catch (and for digging into the code). We have gone through the code and changed every instance of LAI to dEVI

3. On line 305 I think it's important to use language like the data provided much stronger support for model D. The AIC scores aside, there is sound biological reasoning behind the relationships and different data set or structure could more strongly support these other structures, such as. What I'm trying to get at is the nuance of the system and the model of the system - what we infer and predict are affected by the data we collect and how we choose to do the collection and then the variables we choose to calculate to assess how we think the system works.

We agree and now write “..and the data provide much stronger support for hypothesis D (apparent competition) as the top model (Table 1). “. As mentioned above, we also now mention that the models with the same number of parameters are much less likely to be supported (which I think addresses your point about how our design and winning model is a reasonable approximation of how the system functions)

4. Figure 3: suggest making error bars black and removing minor grid lines

Agreed.

5. Figure 4 is showing bivariate, not univariate, relationships.

- Adding predicted lines to these panels would be a nice addition
- Recommend removing the minor grid lines,
- anchoring x-axis of B at 0
- include one more tick value near the origin in A

Agreed, plus we added the plots that Reviewer 3 requested.

6. Discussion: Organizationally, most of the first paragraph could be better situated as the concluding paragraph. My recommendation is to reverse the paragraph ordering to some degree so that it starts with putting the specific results from this study back into broader contexts and expands from there back to the global stage.

Thanks for this comment. We feel this is largely a stylistic issue, and respectfully request to maintain the current organizational structure.

Typos and grammar

- L123-124: parenthetical abbreviation for Northwest Territory but not other provinces
- L224: what does MCP stand for?

Minimum convex polygon, and we think this clarification will help with the previous point re. what the 598,000 km² refers to.

Referee: 3

Comments to the Author(s)

This MS describes a coarse comparison of green vegetation abundance, forest disturbance level, moose density, wolf density, and estimated rate of woodland caribou population change, to test 5 general hypotheses that might be expected to apply when extensive forest clearing occurs. This question is highly relevant in both a general sense to community and ecosystem ecologists and in a conservation sense because woodland caribou are declining across much of their boreal forest range in Canada. Data for wildlife management units were pulled across a wide spatial range. Path analysis was used to discriminate among competing models, with results clearly favoring apparent competition between moose and caribou, due to elevated levels of wolf predation, as the most plausible model/hypothesis. The paper has many strengths, foremost being its application of data from multiple spatial locations, yielding wide variation in disturbance levels. Since many of the alternative models involve complex causal networks, path analysis is clearly the best (and perhaps only) way to treat such snapshot data. The presentation is clear and well supported by accompanying tables and figures.

We thank the reviewer for these thoughtful comments.

I have only 1 major complaint. Given the small number of data points that go into a crucial analysis, I think readers ought to see all the data displayed in a table and additional subplots added to display the visual evidence the crucial relationships that are responsible for rejecting alternate models. For example, there seems little support for the effect of disturbance per se on caribou lambda. Great - let's see it, so we can assess whether outliers, nonlinearities, or other data features cloud the simple statistical inference.

We have now included additional subplots in the main text in Fig 4. We also provide a link to the raw data table explicitly in the results section (L 152). We chose not to print this data table in the main text because it requires only one mouse click to access in GitHub, it would lengthen the MS with little new information (perhaps redundant with the expanded subplots), and we now number the WSUs on the map so that they can be easily cross-referenced with the raw data, plus the data set has been expanded slightly.

I also think that eutrophication is perhaps a misleading term, because enrichment is not usually interpreted as being synonymous with forest clearing. This semantic issue will no doubt irritate many aquatic ecologists and is not necessary for the paper to be of major importance. I would hate to see its value lost in a silly debate about terminology.

Thank you for this comment. We have debated this ourselves, but note that there are numerous instances in the terrestrial literature where the term eutrophication is used (eg Smith et al. 1999, Hautier et al. 2014).

John Fryxell
University of Guelph

Appendix 3 update

We updated the calibration of moose density estimates derived from camera trap data to aerial survey estimates in Appendix 3 to include additional data. Initially the correction factor was approximately 0.406 times the camera density estimate, which has now been updated to 0.478. This update occurred because more data became available for a separate manuscript.

Literature cited

*Hautier, Yann, Eric W. Seabloom, Elizabeth T. Borer, Peter B. Adler, W. Stanley Harpole, Helmut Hillebrand, Eric M. Lind et al. "Eutrophication weakens stabilizing effects of diversity in natural grasslands." *Nature* 508, no. 7497 (2014): 521-525.*

*Smith, Val H., G. David Tilman, and Jeffery C. Nekola. "Eutrophication: impacts of excess nutrient inputs on freshwater, marine, and terrestrial ecosystems." *Environmental pollution* 100, no. 1-3 (1999): 179-196.*

*Zheng, Zhong, Yongnian Zeng, Songnian Li, and Wei Huang. "A new burn severity index based on land surface temperature and enhanced vegetation index." *International journal of applied earth observation and geoinformation* 45 (2016): 84-94.*

Appendix B

12 December 2020

Dear Dr Sasha Dall and the Proceedings Editorial Board,

Thank you for the rapid turnaround of our manuscript entitled “Trophic consequences of terrestrial eutrophication for a threatened ungulate”.

We have implemented the one change requested by the reviewer, by specifying that the LANDSAT was interpreted using imagery from 2008 to 2010 (Line 174). We very much appreciate this attention to detail.

Upon submission, we were informed that our manuscript was roughly 0.2 pages over the limit, so we made a number of minor edits to improve concision to reduce the word count. We moved some details on the moose surveys to the ESM (Line 202 in the tracked changes document).

Also, to the caption of figure 5, we added a point of clarification: “The dashed line represents a link estimated as part of separate analyses (figure 3; Appendix 2).”, and added this dashed line to the figure. Similarly, for the figure 2 caption, we added: “For C, D, and F, the link from habitat alteration to vegetation was estimated as part of separate analyses (figure 3; Appendix 2).”

These edits are shown using track changes in the accompanying files.

On behalf the authors, we thank you once again,

Dr. Robert Serrouya.